# Prospects of Using Machine Learning and Diamond Nanosensing for High Sensitivity SARS-CoV-2 Diagnosis

**Shahzad Ahmad Qureshi** [1,2,*] , **Haroon Aman** [3,4] **and Romana Schirhagl** [5]

1   Department of Computer and Information Sciences, Pakistan Institute of Engineering and Applied Sciences (PIEAS), Islamabad 45650, Pakistan
2   Centre for Mathematical Sciences, Pakistan Institute of Engineering and Applied Sciences (PIEAS), Islamabad 45650, Pakistan
3   School of Mathematics and Physics, The University of Queensland, St. Lucia, QLD 4072, Australia
4   National Institute of Lasers and Optronics College, Pakistan Institute of Engineering and Applied Sciences (PIEAS), Islamabad 45650, Pakistan
5   University Medical Center Groningen, University of Groningen, Antonius Deusinglaan 1, 9713 AW Groningen, The Netherlands
*   Correspondence: drsaqureshi@pieas.edu.pk

**Abstract:** The worldwide death toll claimed by Acute Respiratory Syndrome Coronavirus Disease 2019 (SARS-CoV), including its prevailed variants, is 6,812,785 (worldometer.com accessed on 14 March 2023). Rapid, reliable, cost-effective, and accurate diagnostic procedures are required to manage pandemics. In this regard, we bring attention to quantum spin magnetic resonance detection using fluorescent nanodiamonds for biosensing, ensuring the benefits of artificial intelligence-based biosensor design on an individual patient level for disease prediction and data interpretation. We compile the relevant literature regarding fluorescent nanodiamonds-based SARS-CoV-2 detection along with a short description of viral proliferation and incubation in the cells. We also propose a potentially effective strategy for artificial intelligence-enhanced SARS-CoV-2 biosensing. A concise overview of the implementation of artificial intelligence algorithms with diamond magnetic nanosensing is included, covering this roadmap's benefits, challenges, and prospects. Some mutations are alpha, beta, gamma, delta, and Omicron with possible symptoms, viz. runny nose, fever, sore throat, diarrhea, and difficulty breathing accompanied by severe body pain. The recommended strategy would deliver reliable and improved diagnostics against possible threats due to SARS-CoV mutations, including possible pathogens in the future.

**Keywords:** artificial intelligence; machine learning; deep learning; fluorescent nanodiamonds; spin relaxometry; COVID-19 biosensing

## 1. Introduction

Severe Acute Respiratory Syndrome-2 (SARS-CoV-2) is a subclass of coronavirus infections known due to its widespread across almost all countries around the globe. This virus, with 85% genetic similarity with COVID-19, has a higher infection rate and severe symptoms [1,2]. SARS-CoV-2 symptoms include fever, cough, and breathing issues. The virus is dangerous for infants as well as adults. It is estimated using simplified assumptions that the virus can spread in a thousand individuals within one month and a million in just two months [3]. Due to person-to-person variation in the virus severity and incubation period, two weeks' quarantine period is recommended. Extensive research has been conducted to develop effective vaccines for treatment and diagnosis. Examples of Food and Drug Administration (FDA)-approved vaccines are Pfizer, Cansino, AstraZeneca, and Sinovac. It is noteworthy that AstraZeneca is widely accepted and regulated in most countries. Since 2019, the coronavirus pandemic has had profound social and economic impacts worldwide, causing inflation, economic slowdown, and unemployment. Therefore, the need to develop better

strategies for healthcare management is realized to tackle a similar pandemic outbreak in the future. According to the World Health Organization (WHO), currently, the circulating variants of concern across the United States are the sub-variants of Omicron viz. XBB.1.5, BQ.1, and BQ.1.1. A recent study investigated the substantially higher transmissibility of XBB.1.5 due to the strong binding affinity of human angiotensin-converting enzyme 2 (hACE2) compared to BQ.1 and BQ.1.1 and lower opposition against the immune system [4].

Mostly, the infection occurs through the nose/mouth by contaminated aerosol inhalation, which can affect healthy subjects even 2 m apart. The virus entry in cells is facilitated by angiotensin-converting enzyme 2 (ACE2) receptors available in various cells in the heart, lungs, and blood vessels in the respiratory tract [5]. Initially, the virus affects the regular functions of ACE2 cells that play a critical role in oxygen supply to cells and tissues. Before cell penetration, the SARS-CoV-2 surface spike protein, coated with glycans, strongly attaches to ACE2 receptors. These glycans also provide camouflage from the body's immune system. The spike protein can be decomposed into two subunits, S1 and S2, where S1 exhibits receptor binding and is susceptible to mutations enabling stronger binding with the ACE2 receptor. The S2 subunit actuates the viral fusion with the target cell membrane. To initiate the cell entry, the virus utilizes the required protease enzymes (either Cathepsin-L or transmembrane serine protease 2 (TMPRSS2)) from the host cell [6]. TMPRSS2 is prevalent in the upper and lower respiratory tract, expressed by endothelial cells, and is considered a strong mediator. Before the cell invasion, the TMPRSS2 breaks the S2 subunit and pulls the virus closer to the host cell. Later, the spike protein folds itself and fuses with the cell membrane. The envelope and membrane proteins accommodate the viral accumulation and germination mechanism through the cell membrane after the injection of viral ribonucleic acid (RNA) into the cell. In later stages, the viral genetic material replicates and forms bubble-shaped organelles within the cells. Once the virus grows sufficiently, it leaves the host cell and spreads further to infect the neighboring cells. A schematic view of the SARS-CoV-2 structure and its entry into the cell is shown in Figure 1.

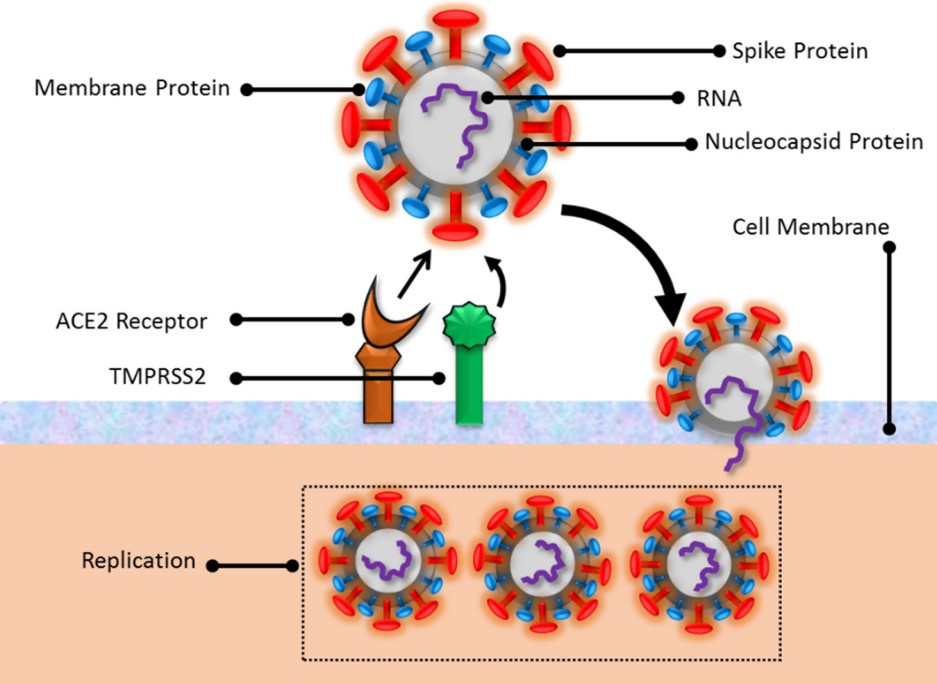

**Figure 1.** An illustration of SARS-CoV-2 entry in the cells.

Biosensors can be classified into different categories such as optical, electrochemical, piezoelectric, microbial, immunological, and enzyme-based biosensors. Each of these available options is suitable for a specific class of diseases such as infections (human immunodeficiency virus (HIV), COVID-19, malaria), cancer (lung, breast, prostate), neu-

rological disorders (Parkinson, Alzheimer, epilepsy), genetic diseases (sickle cell anemia, thalassemia, cystic fibrosis), and respiratory disease. Optical biosensors rely on the absorption and emission of the target molecule for detection. In optical detection, the common choice for sensing material is a fluorescent dye (Rhodamine G6, Fluorescein, Cynanine, Alexa fluora), which allows high quantum yield and good stability for infectious diseases, cancer, neurological diseases, autoimmune disease, and genetic diseases [7]. The other alternatives are quantum dots which are cadmium, indium, or lead-based selenide and sulfide compounds [8]. The precise control over quantum dots' particle size and emission wavelength enables imaging and detection of cancer, HIV, influenza, and cardiovascular diseases. Electrochemical biosensors employ a sensing platform as a microchip that detects the electrical signal (current, voltage) near the electrode due to chemical reaction [9], mostly utilized for infectious diseases, diabetes, and cancer. Microbial and enzyme-based biosensors show high sensitivity and can detect multiple target species [10]. These techniques utilize living organisms such as bacteria, yeast, or algae, which are genetically modified to express physiological and metabolic activity in the presence of specific analytes (deoxyribonucleic acid (DNA), protein, antibodies) through chemical reactions. The enzyme biosensor facilitates the target biochemical reaction (immobilized enzymes) to produce a detectable change in the microorganism's optical, electrical, and pH value [11]. These biosensors are useful for diagnosing diabetes, cardiac injury, and kidney and liver diseases. A flowchart description of the biosensors with disease application is shown in Figure 2.

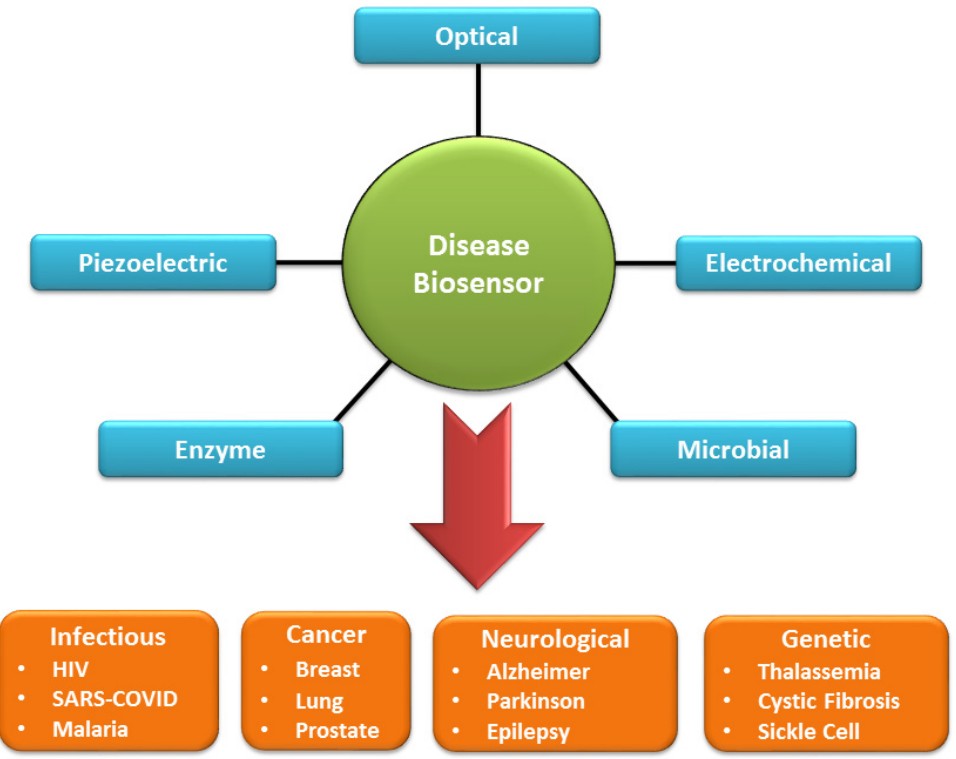

**Figure 2.** A flowchart depiction of general types of biosensors and the variety of target diseases.

## 2. Virus Load and Capability of Reverse Transcription–Polymerase Chain Reaction (RT-PCR)

A recent study has quantitatively reported the growth rate and biological mass of SARS-CoV-2 virions in humans, derived from experiments conducted on monkeys. It is estimated that an infected person at the peak infection could carry $10^9$–$10^{11}$ virions corresponding to a mass of 1–100 micrograms [12]. The study also estimated the possible number of viral RNA copies in different tissue lines viz. nasal mucosa ($10^6$–$10^8$), trachea ($10^6$–$10^9$), tonsil ($10^6$–$10^9$), lungs ($10^9$–$10^{11}$), and the digestive system ($10^3$–$10^7$) [5]. It is inferred that a patient possibly carries $10^4$–$10^6$ infected cells at a given time, leading

to 10 infectious units per cell. The virus can multiply into the host cells in 10–15 min, releasing secondary progeny within 7–8 h (yielding 600–700 malicious units per cell) [3]. The infection prevails for 3–4 days, whereas the associated disease lasts about 2–6 weeks. These data are critical to access the SARS-CoV-2 growth and immune system response to develop vaccines and diagnostic tools.

The current standard diagnostic method is RT-PCR, which relies upon a nasopharyngeal swab test enabling a detection limit of approximately 100 copies of virus RNA in the best-case. However, increasing the detection limit by ten is expected to increase the false negative rate by 13% [13], where the latter drastically changes during the disease. Alternatively, the rapid antigen test is based on immunochromatography using a test strip targeting the virus nucleocapsid protein. This method is more advantageous than RT-PCR due to its fast turnaround time (~20 min) and easy availability of low-cost personal test kits. The testing can now be conducted in laboratories, hospitals, and by individuals without specialized knowledge. However, it is recommended that patients having initial symptoms of SARS-CoV-2 should undergo a rapid antigen test due to its lower accuracy and higher false negative rate approaching 50%. Consequently, a positive verification of SARS-CoV-2 infection is conducted using the RT-PCR test [2].

## 3. Artificial Intelligence in Biosensor Synthesis

Artificial intelligence (AI) is a human intelligence simulation using computer systems. It can develop expert systems for task automation and accurate decision-making and create new products and services available [14,15]. AI aims to make the computer system capable of using new inputs by transforming experiences into expertise. Machine learning (ML) and deep learning (DL) are the two key enablers of AI. ML can adapt automatically with the least human involvement. DL is a subset of ML that mimics the human brain for a learning process powered by an automatic feature extraction strategy. ML and DL algorithms can be based on supervised, unsupervised, or reinforcement learning types. In supervised learning, the algorithm is trained on data (input) annotated by experts, where the target values (output or label) are provided. The resulting optimized model then makes predictions on new or unlabeled input. Unsupervised learning does not use labeled input. It identifies hierarchy patterns in the data and groups similar data together, forming clusters. Reinforcement learning involves training an algorithm to make decisions based on feedback from its environment. DL algorithms comprise layers of interconnected nodes, or neurons, that process and transmit information.

Artificial intelligence recently appeared in the spotlight as an innovative tool for developing SARS-CoV-2-related drugs and predicting the molecular interactions between spike protein and ACE2 receptors [16]. Deep learning models, an important sub-variant of AI, predict suitable candidates from a database of natural compounds/proteins chemically interactive with spike and nucleocapsid proteins, thereby enhancing the accuracy in clinical trials. One of the available databases is the Zinc-database, a free library of FDA-approved chemical compounds and anti-viral drugs that undergo a collective strategy of supervised machine learning, molecular docking, and virtual screening. This helps to evaluate drugs that exhibit strong binding affinity towards nucleocapsid protein, spike protein, and 2′-o-ribose methyltransferase [17]. With the available chemical structure of the latter and the ACE2 receptor binding in the catalytic domain, a 3D homolographic model is generated. Deep learning (DL)-based chemical selection results have been verified via statistical analysis (Naïve-Bayes classifier) to generate a binding energy ranking of anti-SARS-CoV-2 drug candidates.

Additionally, the development of SARS-CoV-2 vaccines (Atazanavir, Efavirenzand, and Ritonavir) has been conducted through genomic sequence identification by using deep learning (convolution neural networks) [18]. These innovations require a reliable source of the chemical database, immune informatics, AI algorithms, chemical structural details of target proteins, and reverse vaccinology [19]. A similar study engaged a DL-based predictive model for investigating bioactive molecules, potentially 3C-like protease inhibitors

using a de novo design [20]. Further, transfer learning enabled specific protease feature recognition, whereas reinforcement learning helped to optimize the desired characteristics of novel molecules. MolAICal, an AI-powered software for designing COVID drugs, inspects the preexisting FDA-approved compounds. By using this software, genetic and DL simulations can be performed for model training using the ZINC database and other medical healthcare resources for a set of predefined rules such as molecular docking and synthetic accessibility [21]. The results are promising for SARS-CoV-2 protein binding with various novel ligands.

Another effort towards designing and developing AI-driven SARS-CoV-2 drugs is the CoronaDB-AI database of natural compounds, proteins, and amino acids, which can train models to rapidly discover effective drugs and target enzyme inhibitors [16,22]. ML-based molecular docking is a primary step toward discovering virtual drugs, demanding the analysis of the molecular structure and chemical bonding of the target molecules. The derived molecules using this technique fit well over the S1-subunit binding spot of SARS-CoV-2. The binding affinity of spike protein with the ACE2 receptor and its trend due to emerging point mutations has been studied using molecular dynamics and free energy perturbations. It was found that N501Y, E484K, and K417N mutations are susceptible to stronger binding with ACE2 receptors [23]. It is noteworthy that some of the AI-developed compounds were also reactive to HIV and other respiratory infections.

## 4. AI Optimization in Nanosensors and Nanomedicine

Optimization of the chemical properties of nanobiosensors can be established by using sequential modeling based on the Bayesian optimization algorithm. This method employs the basic material properties (electrical, optical, and chemical structures). It predicts the optimal conditions that fit the desired parameters for actual experimentation (time, temperature, concentration) by iteratively minimizing an error function. The procedure for the synthesis of biosensors can be extended to a multi-dimensional solution in the chemical space, fusing several fundamental features or key elements. The outcome generates a list of possible reactive candidates for a specific biosensing application. Additionally, the inverse design of biosensors using predefined nanomaterials allows for the formation of a programmable biosensor to achieve the desired optical and chemical characteristics with enhanced sensitivity. In this regard, machine learning algorithms have been used to synthesize different types of nanoparticles (metallic, polymers, and carbon-based), some of which are also useful in nanomedicine [24–28]. A flowchart representing the general principle of using AI techniques to predict biosensors' characteristics is illustrated in Figure 3.

Nanomedicine employs nanomaterials for various imaging, diagnosis, and therapeutic purposes to deliver effective treatment options in healthcare. Generally, hybrid magnetic biosensors are used to detect paramagnetic molecular species, enabling opportunities such as the intracellular identification of free radicals, proteins, and enzymes [29]. There are several ways where biosensors can effectively enhance the diagnosis, drug delivery, and targeted treatment. A high signal-to-noise ratio is desirable in diagnosing tumors, where magnetic nanoparticles carefully monitored using biosensors can be used as Magnetic Resonance Imaging (MRI) contrast-enhancing agents [30]. The blood–brain barrier restricts the proliferation of nanomedicine in glial cells (brain). Here nanodiamonds and chemotherapy drugs have proved to be efficient drug delivery agents [31,32]. Under the influence of precisely tailored external magnetic fields, magnetic nanoparticles can introduce localized magnetic hyperthermia for targeted therapy and treatment of breast and lung cancer [29,33]. The utility of magnetic biosensors in nanomedicine is illustrated in Figure 4.

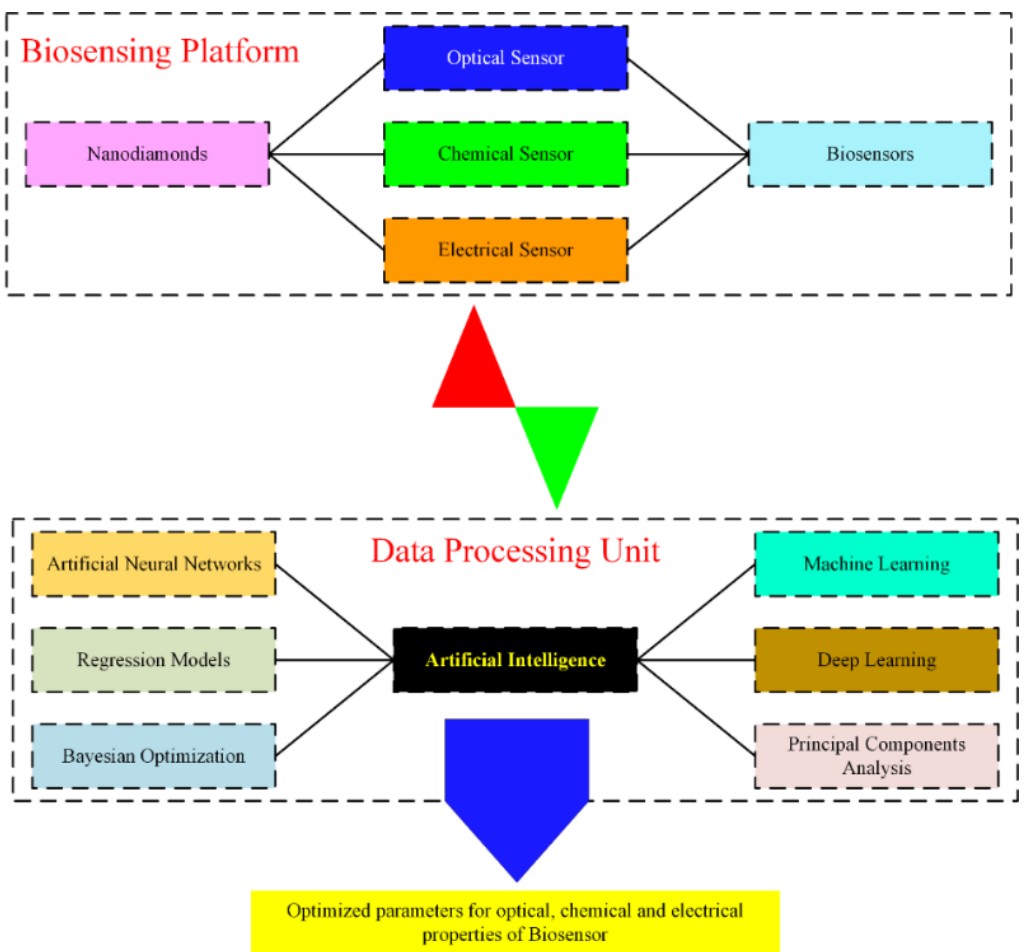

**Figure 3.** Flow chart of the implementation of AI algorithms to predict desired characteristics of a biosensor.

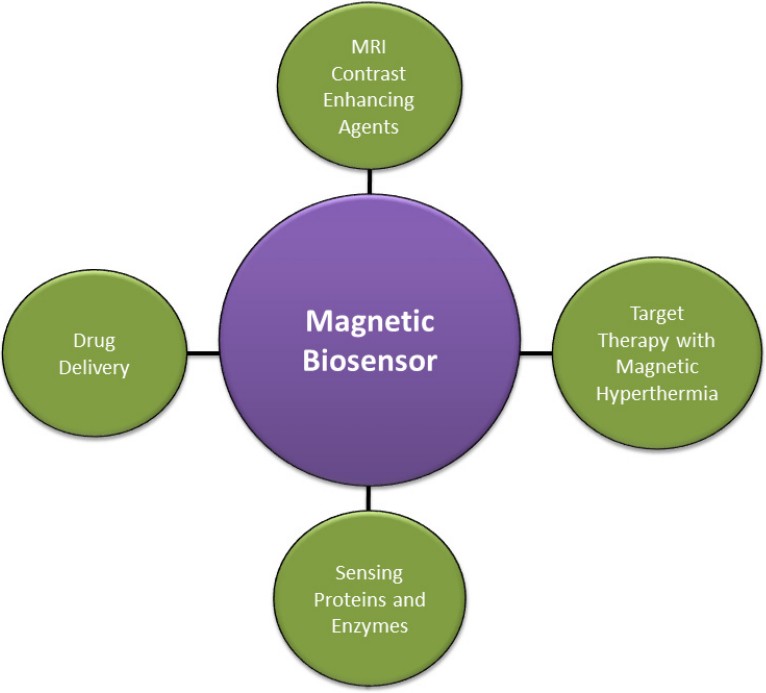

**Figure 4.** Applications of magnetic biosensors in nanomedicines.

## 5. Fluorescent Nanodiamond Role in SARS-CoV-2 Diagnosis

Nitrogen-vacancy centers (NV-centers) in fluorescent nanodiamonds (FNDs) have been extensively used as quantum nanosensors for biosensing and imaging. Due to high photostability, biocompatibility, low toxicity, and surface conjugation with various functional groups, FNDs are ideal candidates for conducting in vivo nanothermometry and magnetic microscopy [34]. The electronic spin of NV-centers allows optical manipulation and readout for high-sensitivity measurements enabling nanoscale spatial resolution and nanotesla sensitivity for stationary and time-dependent fields [35]. By employing the NV spin relaxometry protocols, magnetic field sensitivity can be sufficiently enhanced to detect single and compound biomolecules such as protein, RNA, and DNA [36–41]. The search for high-sensitivity optical diagnosis revealed the successful application of NV-centers in FNDs to detect SARS-CoV-2-related pathogens [36,42]. The novel technique employs a microfluidic device carrying surface-functionalized FNDs as nano-biosensors, where the SARS-CoV-2 RNA extracted from patients is loaded [36]. The FNDs are surface-coated with cationic polyethyleneimine polymer (PEI) so that the SARS-CoV-2 complementary DNA (cDNA), derived earlier from the virus RNA, can be adsorbed. CDNA is chemically bound with Gadolinium ($Gd^{3+}$) complex molecules and shows a detachment from FNDs in the presence of virus RNA. The separation of $Gd^{3+}$ magnetic molecules from the FND surface can be detected and translated into the change in fluorescence intensity by using magnetic microscopy. The optical measurements can be quantitatively evaluated to determine the change in the $T_1$ relaxation time of the NV-spin due to the remaining $Gd^{3+}$ magnetic noise. The outcomes of experimental results supported with simulations disclose that a few hundred copies of virus RNA can be detected in approximately one second with a false negative rate (FNR) of <1%, significantly lower than that of the RT-PCR test. A graphical illustration of the SARS-CoV-2 detection using FNDs is shown in Figure 5. This experiment demonstrates rapid optical biosensing of the SARS-CoV-2 pathogen with high accuracy and substantially lower FNR.

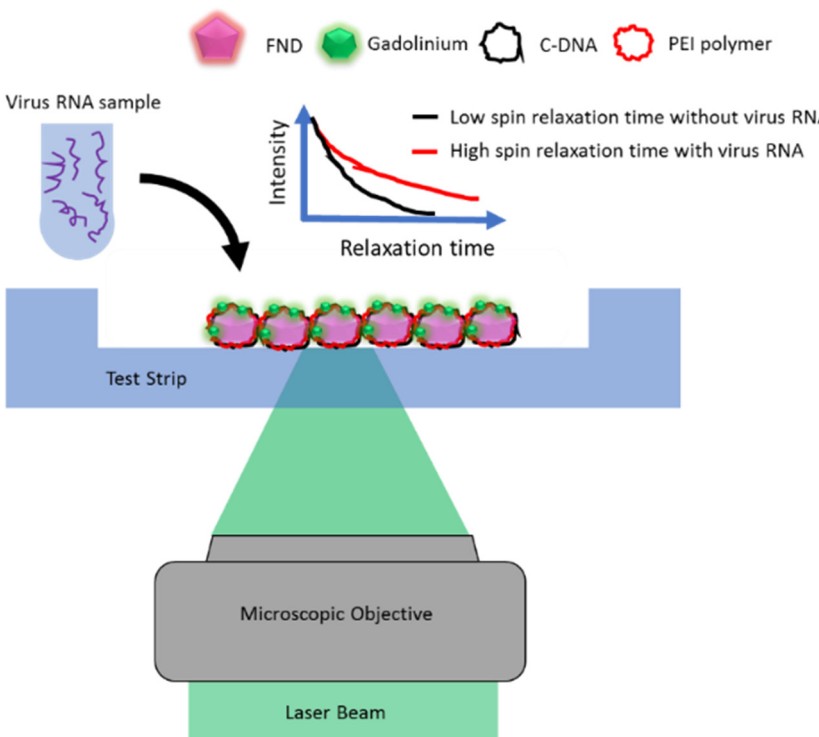

**Figure 5.** Schematic of the SARS-CoV-2 RNA detection using a microfluidic device based on the NV spin relaxometry.

Recent work in this domain highlighted the detection of nucleocapsid and spike protein of different SARS-CoV-2 mutations (alpha, beta, delta, wild-type and Omicron) using Spin-Enhanced Lateral Flow Immunoassay (SELFIA) [42]. The FNDs were coated with SARS-CoV-2 antibodies (S44F, S8-IgG) using bovine serum albumin and phosphate-buffered saline solution and were then subjected to the SELFIA platform. The FNDs exhibit strong binding affinity with the virus antigens leading to a highly sensitive and accurate diagnosis. The SELFIA scheme relies upon the magnetic modulation of NV fluorescence, which is collected using an optical microscope. As mentioned above, the results show that the S44F can effectively detect spike protein antigens for all the mutations. Using a competitive SELFIA assay, the sensitivity can be enhanced by ~50 fold compared to direct SELFIA, pursuing a detection limit of (0.77–1.94) ng/mL, equivalent to (4.4–11) pM concentration. The strategy reported here enables accurate diagnosis and screening at an early stage of infection. The NV-spin relaxometry offers a robust and reliable optical biosensing technique for sensing paramagnetic species with additional benefits such as detecting radicals, pH levels, and redox states in the biological environment [43,44]. The spin relaxation time lowers under the influence of magnetic noise due to quantum decoherence, where typically, it falls within the range of a few hundred microseconds. NV spin relaxometry offers a versatile mode of room-temperature optical biosensing, utilizing low sample volumes as in the case of single mitochondria and a wide range of other intracellular biosensing applications [37,45].

Magnetic nanoparticles, used as contrast-enhancing agents in magnetic resonance imaging [30,46–48], strongly affect the measured $T_1$ relaxation times, enabling us to visualize labeled intracellular organelles. Commercially available FNDs mostly suffer from lower coherence time due to surface defects and active paramagnetic impurities such as $^{13}C$ and $^{15}N$ [34]. The sensitivity of spin relaxometry can be enhanced significantly (100 times) by involving magnetic nanoparticles (Gadopentetate dimeglumine) attached to the tip of the atomic force microscope, which is brought into the vicinity of NV centers [49]. This strategy evolves faster relaxation in NV spins, enabling possibilities for sensing paramagnetic spin in proteins (Hemoglobin, Myoglobin, Cytochrome, Ferredoxin) and enzymes (Cytochrome-P450, Superoxide dismutase, Xanthine).

## 6. Limitations and Challenges for Virus Detection

Relaxometry-based quantum nanosensing faces several challenges due to variations in the response of individual nanosensors and their surface impurities. The NV photoluminescence intensity detected using commercially available FNDs shows strong fluctuations because of the wide size distribution. Therefore, calibration of individual FNDs as nanosensors is necessary before the actual measurements. The magnetic signal acquired from a large FND is generally interpreted as an average response of all the localized quantum defects, where the sensitivity depends inversely on the detected photons. Therefore, using FNDs with high NV concentrations is desirable to acquire a high count-rate and adequate signal-to-noise ratio.

Moreover, the optical excitation and readout must be taken repeatedly for a fixed time delay to reduce the optical noise. There is also a restriction on the distance between the nanosensor and the target biomolecule, which confines this technique toward detecting stationary targets. Further, the complexity of the problem can also be affected if the target species contains magnetic impurities with unknown concentrations. In conclusion, quantum spin relaxometry appears as a versatile technique that offers a broad range of biosensing applications, including magnetic molecules [50–52], chemical redox reactions [53], localized chemical reactions [54], electrolytes [55], pH measurement [56], chemical potential [57], protein binding [58], and free radicals detection in biochemical reactions [59–62].

## 7. AI Integration with FND Biosensing

There are several ways that AI can enhance FND-based biosensing in biological species. Most of the FND biosensing applications utilize the photoluminescence of NV centers as a

nanoprobe. The sensitivity of magnetic field sensing depends on the inverse square root of the detected photons. Hence, the FNDs with high integrated count rates are desirable to achieve the required sensitivity and signal-to-noise ratio. Unfortunately, the commercially available FNDs have shown wide distribution in size (20–25%), brightness, low quantum yield (5–20%), and non-uniformity in shape, due to which only 20% of FNDs have been found suitable for practical applications [63]. In this regard, AI algorithms (ML, DL) can potentially be used to synthesize high-brightness and high-quantum yield FNDs with predefined orientations of NV centers [64]. The idea has been implemented to fabricate high-quantum yield carbon dots and carbon nanotubes [65,66]. In the next stage, the FND surface conjugation and chemical interaction with biomolecules (proteins, enzymes, antigens, intracellular organelles, therapeutic nanomedicines) can also be predicted for preparing different combinations of test samples, including the concentration of precursors and their reaction time [67–69]. Once the biosensor is merged with biological media, the optical signal acquired from FNDs through the experimental procedure can also be improved and supervised for biomarker identification and any unpredictable anomalies in real-time [70,71]. Additionally, the experimental data can be processed, interpreted, and reconstructed to remove undesired autofluorescence background [72,73]. The process flow of AI-integration through ML for SARS-CoV-2 is illustrated in Figure 6.

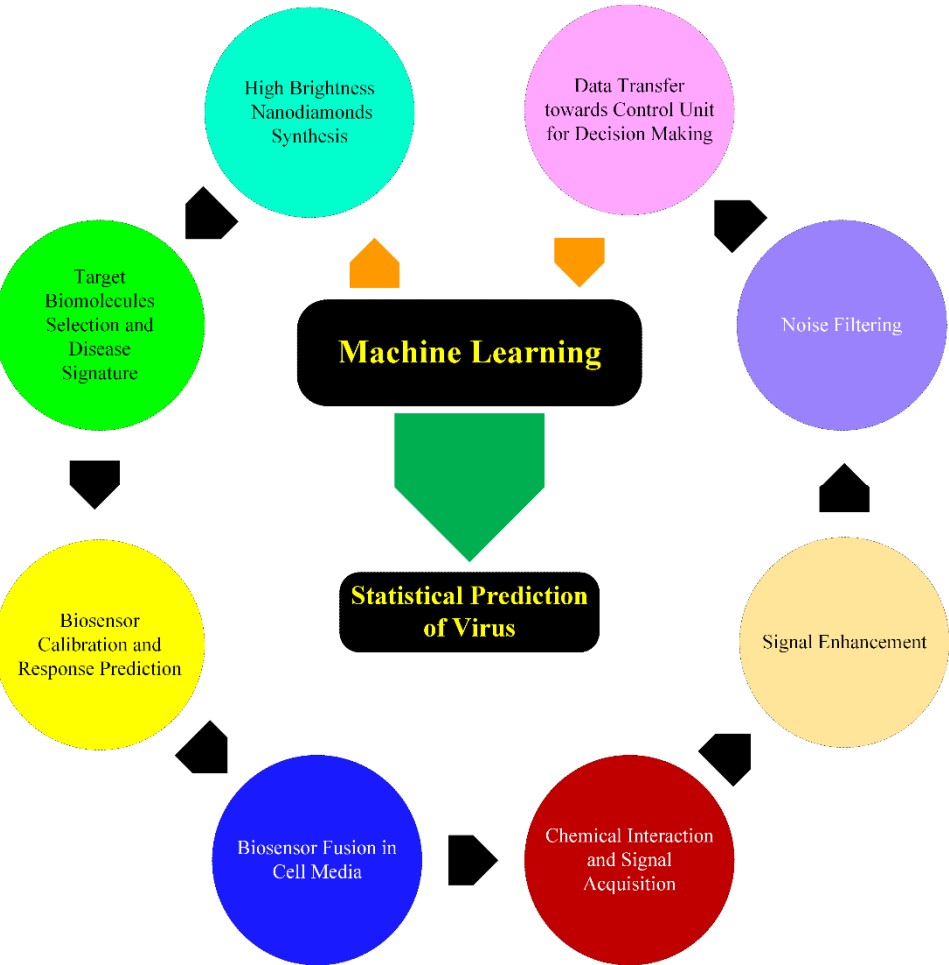

**Figure 6.** Data flow diagram for the proposed model of AI-integration with FND biosensing for SARS-CoV-2 prediction.

Recent reports have shown a few experimental demonstrations of AI-based enhancement of FNDs' nano-biosensing capabilities. The in vivo tracking of fluorescent biomarkers is critical for the screening and testing of the uptake and efficacy of novel medicines. The fluorescent biomarkers are optically excited, and their fluorescence emission is detected, typically mixed

with an autofluorescence background from the cellular media. The background autofluorescence can restrain the detection of a biomarker available at low concentrations. Using the inverse problem for the separation of biomarker signal from the background artificial neural network (ANN) algorithm has been tested to determine the concentration of 100 nm FNDs from the samples of human urine [74]. The technique combines Raman and fluorescence spectroscopy to acquire NV emission within the 640–800 nm band. The method is applicable for 1–20 mg/L solutions with an accuracy of 0.3 mg/L. A similar study reported the detection of low concentrations (3 μg/mL) of FNDs from chicken egg white [75]. The presence of magnetically labeled cancer cells in human lung and breast tissues has been realized using DL-based algorithms, which can reconstruct the density of magnetic nanoparticles from optical images [76]. The technique is used for immunomagnetic microscopy of the Epidermal Growth Factor Receptor. It can deliver a spatial resolution of ~20 μm with the additional benefits of background noise correction and tumor growth monitoring. AI-based optimization of concentration and treatment time in combinational drug therapy using different drugs in clinical research has direct employment in diseases such as diabetes, HIV, and cancer [77]. Accounting for safety, the production of nanomedicine with high efficacy and minimal side effects is a challenging task at a large scale. AI-based feedback control system with FND-conjugated nanomedicine has been tested on different breast cancer cells (MCF7, DA-MB-231, BT20) [78]. The study has revealed the successful implementation of AI in optimizing drug delivery with the required viability. A tabulated summary of the AI-based SARS-CoV-2 diagnosis techniques is shown in Table 1.

The enhancement in the speed of FND-based magnetic sensing can directly contribute towards the visualization and measurements of dynamic intracellular reactions such as metabolic heat generation, interneuron signaling, and chemical redox. The sequential Bayesian approach combined with statistical analysis has shown enhancement in the speed of magnetic field reconstruction by an order of magnitude [79]. Another experiment revealed that Gaussian regression could be used to estimate the magnetic field with an accuracy of 2 μT for low fields ~2.2 mT [80]. Finally, we suggest different scenarios under which AI can be integrated with FND biosensing for SARS-CoV-2 detection. This challenging task will require AI algorithms to screen the available database of existing drugs and molecules to select specific biomolecules that exhibit strong binding with the spike protein and surface-enhanced FNDs. Some ongoing projects relating to AI for exploring the drugs which show binding with spike protein are exscalate4cov, benevolent, and atomwise. If chemically bound with magnetic nanoparticles, these selected biomolecules can be directly used for spin relaxometry measurements using microfluidic devices. The procedure will assist in identifying the presence of virus entities at very low concentrations. Overall, the main role of AI will be to develop novel biomolecules or identify the existing ones that fit for FND-based SARS-CoV-2 biosensing.

**Table 1.** AI-based published research work for SARS-CoV-2 diagnosis.

| Ref. | Authors | AI Model | Method Description | Imaging Technique | Prediction Accuracy | Comments/Limitations |
|------|---------|----------|--------------------|-------------------|---------------------|----------------------|
| [81] | Singh et al. | ML | Hybrid Social Group Optimization Algorithm-based feature extraction and Support Vector Machine (SVM) classifier | X-rays | 99.65% | High-class imbalance in the dataset due to a limited number of COVID-19 positive images |
| [82] | Elaziz et al. | ML | Feature selection using an optimization algorithm and classification using k-nearest neighbors (k-NN) classifier | X-rays | 96.09% and 98.09% for datasets 1 and 2, respectively | The class imbalance was present in both datasets (1 and 2) with 216 and 219 COVID-19 positive images respectively; cross-validation of results was not implemented |

**Table 1.** *Cont.*

| Ref. | Authors | AI Model | Method Description | Imaging Technique | Prediction Accuracy | Comments/Limitations |
|---|---|---|---|---|---|---|
| [83] | Biswas et al. | DL | Transfer learning based on an ensemble of visual geometry group (VGG)-16, residual network (ResNet)-50, and Xception architectures | CAT * scans | 98.79% | Stack generalization was used as an alternative to the cross-validation of the prediction model |
| [84] | Jangam et al. | DL | Stacked heterogeneous ensemble classifier of VGG-19, ResNet-101, densely connected convolutional network (DenseNet)-169, and wide residual network (WideResNet)-50-2 | CAT scans | 85.71%, 99%, and 93.5% for datasets 1, 2, and 3, respectively | Training and testing times were high which can be alleviated with parallel computing algorithms using NVIDIA graphics processing unit (gpu)-boost cards |
| [85] | Shankar et al. | DL | Cascaded recurrent neural network (barnacle mating optimization (BMO)-cRNN) using BMO for feature extraction | X-rays | 97.31% | High-class imbalance with instances spread as 27:220 (normal: COVID-19) |
| [86] | Sarki et al. | DL | Transfer learning from scratch by employing VGG-16, Inception V3, and Xception | X-rays | 93.75% (second case) | Limited availability of high-quality COVID-19 public image was the main problem, resulting in lower test images |
| [87] | Mansour et al. | DL | Variational auto-encoders (VAE) for unsupervised learning and classification using Inception V4 for feature extraction (Adagrad technique) | X-rays | 98.7% | Metaheuristic parametric learning strategy may be used to improve the results further |
| [88] | Elmuogy et al. | DL | Transfer learning using worried deep neural network(WDNN) | CAT scans | 99.046% | The algorithms are without cross-validation |
| [89] | Wang et al. | DL | Modified inception (M-inception) model using region of interest (ROI) images | CAT scans | 89.5% | The CT images in training were reported deficient by the authors |
| [90] | Kumar et al. | ML and DL | Feature extraction by ResNet152 with ML classifiers such as k-NN, decision trees, and adaptive boosting | X-rays | 97.7% | Synthetic images used during training with the help of the synthetic minority oversampling technique (SMOTE) |

* Computer Aided Tomography (CAT).

## 8. Conclusions

In this paper, we emphasize the role of FNDs-based quantum nanosensing for sensitive and accurate SARS-CoV-2 diagnosis and any possible pathogenic outbreak due to continuing mutations. The full benefit of FND-based biosensing can be harnessed by assessing the worldwide SARS-CoV-2 database available by WHO and healthcare centers. Machine learning tools can now mimic the chemical interaction between the commercially available natural and synthetic molecules with different classes of spike proteins. In this regard, AI can determine the composition of bio-conjugated FNDs while the efficacy and side effects can be analyzed in clinical trials. These AI-developed biosensors combined with spin relaxometry measurements could have the potential for SARS-CoV-2 detection at extremely low concentrations. In case of a pandemic outbreak, the availability of an accumulated molecular database using simulations and clinical trials would greatly enhance the efficiency of machine learning tools in understanding the biochemical nature of pathogens at the molecular level and the design of customized biosensors. FNDs-based biosensing

currently faces challenges regarding implementation in the complex biological environment such as precise intracellular control on the location of the sensor and signal reproducibility in the presence of strong background auto-fluorescence. The consequences of free radical detection can be further extended to investigate several physiological processes, such as intercellular signaling, cellular metabolism, and biochemical reactions, due to the immune system's response to viral infection.

**Author Contributions:** All authors have equal contributions. All authors have read and agreed to the published version of the manuscript.

**Funding:** This research received no external funding.

**Institutional Review Board Statement:** Not applicable.

**Informed Consent Statement:** Not applicable.

**Data Availability Statement:** Not applicable.

**Conflicts of Interest:** The authors declare no conflict of interest.

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
