# Peer review of "Prospects of Using Machine Learning and Diamond Nanosensing for High Sensitivity SARS-CoV-2 Diagnosis"

_magnetochemistry, doi:10.3390/magnetochemistry9070171_

Round 1
Reviewer 1 Report
In the manuscript: prospects of using machine learning & quantum bio-sensing for high sensitivity SARS Cov-2 diagnosis, the authors bring attention to quantum spin magnetic resonance detect using fluorescent Nano-diamonds for bio-sensing, ensuring the benefits of artificial intelligence-based biosensor design on an individual patient level for disease prediction and data interpretation. This manuscript written well but should enriched with enough figure in some subtitles for explain the viewpoint of authors.
Author Response
Please find attached the response to Reviewer 1 in attached file.

Reviewer 2 Report
In this perspective, the development of a biosensor based on fluorescent nanodiamonds (FND) for sensitive and accurate diagnosis of SARS-Cov-2 enhanced by artificial intelligence is reported. Therefore, the FND-based biosensor can be used through the worldwide SARS-Cov-2 database available by WHO and health centers for detecting the virus of an individual patient to predict future illnesses.
There are however several concerns that I would like the authors to consider.
Major revision:
- The English needs to be corrected and revised, some words in the text are misspelled, such as "labelled", "modelling", "signalling". Please review all text carefully.
- The introduction needs to be redrafted. Some terms are used for the first time without a quick explanation of what they are.
- The basic theme of the article is a quantum biosensor for diagnosis of SARS Cov-2. The addition of a paragraph with some published works for biosensors of some diseases would be interesting to highlight the advantages and importance of a biosensor in this area.
- In the introduction, at the beginning of page 2, the authors cite the acronym "WHO", but do not previously report what it is about. Please identify what this acronym means in the text.
- The authors cite "ACE2" in the introduction, at the beginning of page 2, but this term should be explained more clearly since it is fundamental for the understanding of the article.
- In the topic "2. Virus load and capability of RT-PCR", the authors should detail what the acronym "RT-PCR" means.
- In the topic "4. AI optimization in nanosensors and nanomedicine" more works should be described as biosensors in nanomedicine. The paragraph is very lacking in detail and should be improved.
- In the topic "6. Limitations and challenges for virus detection" the author describes the limitations of virus detection. However, the topic needs more details and approaches. Relaxometry-based quantum nanosensing should be cited for other works involving the detection of viruses or other target substances. There is a lack of quotes and deep work on this topic.
- A figure for each topic of the article should be added. The article is poor in figures or schemes. As it is a perspective, illustrations should be added so that the reader can better understand the idea in the text.
The English needs to be corrected and revised, some words in the text are misspelled. English must be revised by a native speaker.
Author Response
Please find attached the response to Reviewer 2 in attached file.

Reviewer 3 Report
I have some comments regarding the focus, content, and presentation of the papertitled "Prospects of using Machine Learning & Quantum Biosensing for High Sensitivity SARS Cov-2 Diagnosis."
Firstly, it is evident that the majority of the paper revolves around the use of nanodiamond sensors, with limited discussion on other quantum biosensing approaches. This creates a discrepancy between the title and the actual content of the paper. To address this, I recommend either modifying the title to accurately reflect the focus on nanodiamond-based sensing or expanding the scope to encompass a broader range of quantum biosensing techniques for SARS Cov-2 diagnosis.
Additionally, as this is a Perspective article, it would greatly benefit from the inclusion of visual aids to enhance the understanding and clarity of the presented concepts. I suggest incorporating a figure that illustrates the relationship between machine learning and quantum biosensing, showcasing how these two fields intersect in the context of SARS Cov-2 diagnosis. This visual representation would provide readers with a clear overview of the synergistic integration of these approaches.
Furthermore, given the importance of artificial intelligence (AI) in conjunction with quantum biosensing for SARS Cov-2 diagnosis, I recommend organizing the information into a concise and informative table. This table should outline the different AI techniques employed in the field of quantum biosensing for SARS Cov-2 diagnosis, along with corresponding applications, advantages, and limitations. Such a table would greatly assist readers in comprehending the current landscape of AI and quantum biosensing integration.
.
Author Response
Please find attached the response to Reviewer 3 in attached file.

Round 2
Reviewer 2 Report
Dear authors,
The manuscript was corrected and revised as suggested by the reviewer. Thus, the manuscript can be accepted for publication.
Reviewer 3 Report
I am satisfied with the authors' responses and have no further comments.
.